# Intraobserver Assessment of Shear Wave Elastography in Tensor Fasciae Latae and Gluteus Maximus Muscle: The Importance of the Hip Abductor Muscles in Runners Knee Compared to Healthy Controls

**DOI:** 10.3390/jcm11133605

**Published:** 2022-06-22

**Authors:** Andrea S. Klauser, Felix de-Koekkoek, Christoph Schwabl, Christian Fink, Miriam Friede, Robert Csapo

**Affiliations:** 1Radiology Department, Medical University Innsbruck, Anichstrasse 35, 6020 Innsbruck, Austria; andrea.klauser@i-med.ac.at (A.S.K.); felix.de-koekkoek@tirol-kliniken.at (F.d.-K.); 2Gelenkpunkt—Sport- und Gelenkchirurgie Innsbruck, 6020 Innsbruck, Austria; info@gelenkpunkt.com; 3Fachhochschule Kärnten Gemeinnützige Privatstiftung, 9020 Klagenfurt am Wörthersee, Austria; m.friede@fh-kaernten.at; 4Centre for Sport Science and University Sports, University of Vienna, 1010 Wien, Austria; robert.csapo@univie.ac.at

**Keywords:** sonoelastography, iliotibial band syndrome, ultrasonography, runners knee, hip muscles, tensor fasciae latae, gluteus maximus

## Abstract

Background: Iliotibial band syndrome (ITBS) represents one of the most common running related injuries. The pathophysiology is postulated to be caused by excessive ITB tension, impingement and irritation of soft tissues at the lateral femoral epicondyle. However, direct evidence has yet to be found and the multifactorial etiology is under discussion. The purpose was to evaluate stiffness of ITB, gluteus maximus (GM) and tensor fasciae latae (TFL) muscles using shear wave elastography (SWE). Methods: In 14 patients with clinically verified ITBS and 14 healthy controls, three SWE measurements each of ITB, GM and TFL in both legs was performed to determine measurement reliability and between-group and -leg differences. Results: The mean value of ITB was 12.8 m/s with ICC of 0.76, whereas the values measured in the GM were 3.02 m/s with an ICC of 0.87. No statistically significant difference in controls compared to patients were found (*p* = 0.62). The mean value of TFL was 5.42 m/s in healthy participants, compared to 3.89 m/s patients with an ICC of 0.98 (*p* = 0.002). Conclusion: Although SWE showed no difference in ITB stiffness, significant differences for TFL muscle stiffness in runner’s knee was found, suggesting that the hip abductor muscles might play a bigger role in the pathophysiology of ITBS. We aimed to implement baseline values for stiffness assessments and prove reliability for further prospective studies of SWE in runner’s knee.

## 1. Introduction

Iliotibial band syndrome (ITBS) (i.e., runner’s knee) is characterized by lateral knee pain and represents one of the most common running related injuries, affecting up to 10% of all active runners [1]. It is a non-traumatic overuse injury caused by repeated flexion and extension of the knee leading to irritation in the structures around the knee [2]. Orchard et al. [3] described an impingement zone occurring at 30° of knee flexion during foot strike and the early stance phase of running. During this impingement period in the running cycle, eccentric contraction of the tensor fascia latae muscle and of the gluteus maximus muscle causes the leg to decelerate, generating tension in the iliotibial band, leading to irritation of the underlying soft tissues at the level of the lateral femoral epicondyle [4,5]. ITBS is usually diagnosed on the basis of a detailed history and physical examination [6].

However, direct evidence with objective imaging criteria must still be investigated.

Shear wave elastography (SWE) generated by ultrasound is an innovative ultrasound imaging-based technique facilitating the quantitation of tissue elasticity by measuring the propagation velocity of US shear waves in the tissues of interest. Pilot experiments have demonstrated the feasibility of SWE measurements of ITB in 16 healthy volunteers [7].

As a relatively new imaging technique, SWE allows for a quick, non-invasive quantitative assessment of tissue stiffness [8,9,10,11].

In contrast to strain/compression sonoelastography, where mechanical tissue compression is needed, SWE uses an acoustic radiation force pulse sequence to generate shear waves, which propagate perpendicular to the US beam, causing transient displacements.

The distribution of shear wave velocities at each pixel is directly related to the shear modulus, an absolute measure of the tissue’s elastic properties.

Shear-wave elastography is considered to be more objective, quantitative, and reproducible than compression sonoelastography, and as many studies have showed, SWE seems to have a promising role in determining the severity of disease in various musculoskeletal tissues including tendons, muscles, nerves, fascias and ligaments [8,9,10,12,13].

In addition, MRI is also a suitable method to evaluate the ITB and especially the surrounding structures for differential diagnosis. On MRI, the ITB is a hypointense, flat, linear structure in the lateral hip, thigh, and knee. In the absence of pathology, there should be no adjacent edema or significant intrasubstance signal changes. On MRI, the normal ITB will typically be of a thickness of about 1 to 3 mm at the level of the lateral femoral epicondyle, consistent with sonographic findings [1,14,15]. Nevertheless, the thickening of the ITB has been reported inconsistently in both MRI and US, therefore an independent imaging marker is desirable [16].

The aim of the study was to define baseline values and to prove the reliability of three SWE-based measurements each of ITB, TFL and GM and to compare results of patients clinically diagnosed with ITB-syndrome with a healthy control group. We hypothesized that SWE elastic moduli would be significantly higher in participants diagnosed with ITBS.

## 2. Materials and Methods

All participants gave their informed consent for inclusion before they participated in the study. The study was conducted in accordance with the Declaration of Helsinki, and the protocol was approved by the Institutional Review Boards (IRBs) of our institution (EK Nr: 1090/2018).

The dependent variables were the stiffness of the ITB, TFL and GM as measured by SWE.

The independent variable was the presence of ITBS.

### 2.1. Participants

14 patients (mean age 32.6 ± 6.6/m: f = 7:7) with verified ITBS and 14 healthy volunteers (median age 26.1 ± 5.2/m: f = 7:7) with no statistically relevant difference in age were recruited during 2018–2019. ITBS was verified through clinical examination by an experienced orthopedic surgeon and functional tests were used for differential diagnosis including the Noble, Ober and Thomas tests, which were positive in all of the included patients [17,18,19]. In addition, MRI scans were performed to test the presence of edema and exclude other injuries (e.g., lateral meniscal tear) potentially causing similar symptoms [20]. Patients presenting with such secondary injuries were excluded from the study. A history of precedent lower extremity pain or injuries in the last six months as well as severe injuries (e.g., fractures), surgeries at any time, BMI > 30 or physiotherapy because of ITBS in the last 12 months were further criteria for exclusion. Previous medication was no exclusion criteria.

Participants in the control group had to be healthy and physically active (physical activity > 150 min/week) with no history of ITBS (Global Recommendations on Physical Activity for Health, 2009. World Health Organization. Geneva, Switzerland. Available online: http://www.who.int/ncds/prevention/physical-activity/en/ accessed on 13 July 2018). One of the *participants* from the healthy group had a lateral meniscus tear in his dominant leg, which was one of our exclusion criteria. This participant’s data were removed from analysis; therefore, the analysis was conducted on a sample of 14 patients with ITBS and 13 healthy participants.

### 2.2. US Examination

To obtain SWE images, participants lay relaxed supine on an examination bed with their backs slightly raised and knees rested on a support cushion (hip angle 140–150°, knee angle ~90°). US images were obtained in the sagittal and frontal plane, respectively, in both legs in three locations: proximally, above the tensor fasciae latae (2 cm proximal of the greater trochanter of the femur in the direction of the anterior superior iliac spine) and gluteus maximus muscles (4.5 cm proximal of the greater trochanter of the femur in the direction of the highest point on the iliac crest), and distally above the ITB (2 cm proximal of the lateral femoral epicondyle).

An ultrasound SWE system (Aixplorer Supersonic Imagine, Aix-en-Provence, France) with a 50 mm linear array transducer (SL 18-5, Supersonic Imagine, France) was used with settings in the musculoskeletal mode. The frequency was 18 MHz, and the SWE option was penetration mode with an opacity of 85%. The preset was adjusted to a depth of 1 cm for the iliotibial tract and 3 cm for the gluteal muscles with an elastic scale < 600 kPa. The color scale used in the shear modulus (in kPa) showed the lowest values in blue to the highest values in red. The size of regions of interest (ROI) had to be at least 3 × 10 mm in order to cover the ITB and 50 × 30 mm in order to cover the gluteal muscles. The Q-Box™ diameter was defined by the thickness of the ITB and the muscles, respectively. The Q-Box™ was traced manually to include a maximum of the muscles in order to avoid the muscle-tendon junction and fasciae and to downsize it in order to measure the very thin ITB.

During measurements, an ultrasound gel was applied between the skin and the transducer to avoid skin deformation. The midpoint of the transducer was placed perpendicularly on the skin’s surface on the ITB and muscle fibers with a light pressure and then the SWE mode was activated to examine the shear wave modulus [21]. During the acquisition of the SWE mode, the transducer was kept motionless for about 5–8 s [22]. Image quality was closely monitored throughout the measurements. When the color in the ROI was uniform and the structure of the ITB and the muscle fibers visible, the images were frozen and then put on the Q-Box™ to obtain the shear wave modulus from the system and stored for SWE analysis (kPa, m/s) [23]. For this purpose, the probe was held aligned along the long axes of the GM (axial transducer positioning) and TFL (longitudinal transducer positioning) muscle fibers and at the distal ITB (longitudinal transducer positioning), before regions of interest were manually drawn on frozen images (Figure 1, Figure 2 and Figure 3). In the ITB, care was taken to leave out hyperechoic lines due to cortical bone to avoid potential bias. Three measurements were obtained in each location by freezing and unfreezing and acquiring new images each time with consecutive manual ROI placement in order to prove intraobserver reliability. Shear wave velocities and shear moduli were obtained by a single observer with five years of experience in MSK US and SWE in three locations: Two measurements were performed at the hip level, at approximately 50% of the distance between the anterior superior iliac spine and the greater trochanter of the femur to assess the TFL and the GM muscles, followed by a distal measurement of the ITB obtained 2 cm proximally of the level of the lateral femoral epicondyle. The three measurements were used to calculate intra-observer variability and the mean of the three measurements was used for further statistical analysis. Shear wave velocities are given in m/s.

Note that only TFL and GM and not the gluteus medius muscle were measured, because they contribute fibres to the ITB. The gluteus medius muscle has no direct attachment to the ITB, therefore we did not perform measurements [14,24].

### 2.3. Statistical Analysis

The sample size was determined through a priori power analysis (α = 0.05, 1-β = 0.8, dz = 1) based on previously published data of ITB stiffness [7].

For statistical analyses, we used SPSS version 25 (© IBM). Intraclass correlation coefficients (ICC’s) were calculated using a two-way mixed effects models to quantify absolute agreement of measurements [25]. Typical errors of measurement were calculated by dividing standard deviation of (maximum) difference scores by the square root of two [26]. In addition, the minimal detectable change (MDC) was calculated as: 1.96 × standard error of the mean (SEM) × √2 [27].

Differences in tissue stiffness, as reflected by measures of shear wave propagation velocity (m/s) were tested for significance by means of factorial MANOVA’s and a two-way mixed ANOVA considering measures obtained in the ITB, GM and TFL as dependent variables, “leg” (affected/non dominant versus non-affected/dominant) as within- and “group” (patient versus control group) as between-participant factors, respectively. Box’s test was used to test the assumption of homogeneity of co-variances. For significant differences, Pearson’s coefficient was calculated through n^2^-conversion [28].

A *p*-value less than 0.05 was declared as statistically relevant.

## 3. Results

### 3.1. ITB

ITB showed a mean of 13.24 ± 2.24 m/s (coefficient of variation (CV): 16.92%) in healthy participants and failed to reach statistically significant differences in comparison with the mean of the diseased legs (12.36 ± 2.92 m/s (CV 23.62%)), (P = 0.62), (Figure 1). Neither the differences between dominant/non-dominant as well as diseased/non-diseased legs were statistically relevant (*p* > 0.2).

ITB showed a good intra-observer reliability with an ICC of 0.76 (0.63–0.85). SEM ranged from 0.43 to 0.56. MDC ranged from 1.19 to 1.55.

### 3.2. TFL

TFL values were significantly higher in healthy participants with a mean of 5.42 ± 2.25 m/s (CV: 41.51%) compared with a mean of 3.89 ± 1.92 m/s (CV: 49.36%) in diseased legs (*p* = 0.002, r = 0.41), (Figure 2). Intraobserver reliability was excellent with an ICC of 0.98 (0.96-0.99). SEM ranged from 0.43 to 0.75. MDC ranged from 1.19 to 2.08.

### 3.3. GM

GM mean SWE values failed to reach statistically significant differences with a mean of 2.9 ± 0.95 m/s (CV: 32.76%) in healthy participants and a mean of 3.14 ± 1.73 m/s (CV: 55.10%) in diseased legs (*p* = 0.26), (Figure 3). Intraobserver reliability showed an ICC of 0.87 (0.80-0.92). SEM ranged from 0.18 to 0.33. MDC ranged from 0.50 to 0.91. The TFL/GM ratio was significantly higher in the healthy group compared to the patient’s group (*p* = 0.049, r = 0.41), (Figure 4).

## 4. Discussion

We were able to show in our study that there are no statistically significant differences in SWE values between the patient group with ITBS and the healthy control group. These findings are in line with a recent study examining ITBS and hip abductors using SWE, where no statistical differences in ITB SWE values were found in patients with verified ITBS before and after physiotherapy [29]. This was the first study to apply SWE in ITBS, however reliability data were not referenced in detail. Therefore, the goal of this study was to show that baseline SWE values are reproducible and can, therefore, be used in further studies to assess runners knee. The current hypothesis concerning the etiology of the ITBS is that excessive tone in the ITB leads to the compression of underlying fat tissue and consequently, to inflammation and pain [30].

Another study examining the effect of muscle fatigue using EMG in females with ITBS, however, suggests that the hip abductors of patients suffering from ITBS do not show lower maximum strength but demonstrate less resistance to fatigue than those of healthy runners. Therefore, the study suggested implementing a gluteus medius endurance training regime in a runner’s rehabilitation program [31]. In agreement with this study, a more recent study has identified a number of kinematic differences between injured and healthy runners that were consistent across injured subgroups; pelvic drop was found to be the most important predictor variable [32]. This finding might be in line with our results, supporting the hypothesis that muscular deficits (i.e., hip abductor weakness) might lead to improper posture during the stance phase of the gait cycle (i.e., pelvic drop and valgus collapse), resulting in excessive ITB strains and inflammation. Hence, in many cases, appropriate strength training might represent the first line of treatment and should be considered before more invasive treatment options, such as the injection of local steroids or even surgery are contemplated [33]. It would be of interest to test SWE in ITBS patients in order to test the hypothesis that insufficient hip abductor tone is involved in the pathogenesis of the syndrome. In another study applying SWE, the group of Tateuchi et al. [7] used a transversal scanning plane to investigate the effect of angle and moment of hip and knee joints on ITB stiffness. They noted a ceiling effect of measurements obtained in the longitudinal plane. No such ceiling effect was observed in our study, which may be explained by the newer generation equipment used.

### Limitations

Although data collection was standardized to minimize influencing factors, we have several limitations to mention. First, the ITB is a very thin aponeurotic fascia lying above the cortical bone which represents a challenge in the acquisition of SWE measurements. This often results in saturated values (Figure 1). However, we performed three measurements each, in order to get reproducible values, which mainly did not result in saturated values. Nevertheless, it is questionable if this fact was leading to the lack of statistical differences between patients and controls. Future studies in the ITB should consider using a gel pad. Additionally, subjacent/adjacent edema usually observed in patients suffering from ITBS could contribute to lower SWE speed values [34]. We didn´t compare the thickness of the ITB between groups, which is also a limitation of our study. Perhaps there could be a correlation between thickened ITB and lower speed values. Second, measurements obtained in the ITB are influenced by the resting tension of muscles inserting into it. While all study participants were instructed to fully relax the examined extremity during the US examination, it cannot be excluded that involuntary contraction of GM or the TFL muscles may have introduced bias into ITB SWE measurements. Furthermore, we are aware that stiffness values measured in a relaxed state do not necessarily reflect the conditions during physical activity.

A recent study by Besomi et al. [35] showed different values in ITB stiffness, but in contrast to our study, the ITB was measured more proximally, resulting in different values and appearance. We have decided to measure the ITB at the insertion, as this is also where changes also in the b-mode of patients may occur.

A further limitation is that we did not examine a healthy runners group. This refers to healthy participants without ITBS symptoms with a self-reported weekly training volume of at least 20 km. Training may induce changes in the shear modulus and therefore lead to different baseline values.

A recent SWE study provided evidence that the muscles of active runners exhibit an increased stiffness that can be beneficial to their athletic performance [36]. Increased muscle stiffness in runners might lead to false negative SWE values as compared to healthy volunteers. Moreover, another very recent study showed that women with genu varum alignment exhibit higher ITB strain during weight-bearing, which could be related to a higher incidence of ITBS in women [37]. This was unknown to us prior to this study’s subject selection and therefore not evaluated and should be addressed in further studies.

A further limitation is the relatively small study population, which limits statistical power and may have contributed to the differences in the mean values between the study groups. In this study we had good intra-observer reproducibility, but unfortunately no inter-observer reproducibility was evaluated as the measurements were carried out by only one radiologist, which represents a further limitation.

## 5. Conclusions

SWE is an evolving imaging modality recently available on higher frequency ultrasound transducers which enabled us to obtain reliable quantitative measurements of tissue stiffness of ITB, TFL and GM. SWE showed no difference in ITB stiffness, whereas a significant difference for TFL muscle stiffness was found, suggesting that the hip abductor muscles might play a bigger role in the pathophysiology of ITBS. Our study summarizes SWE values with good intraobserver variability, and can serve as a background for further longitudinal studies of stiffness assessments in runners knee.

## Figures and Tables

**Figure 1 jcm-11-03605-f001:**
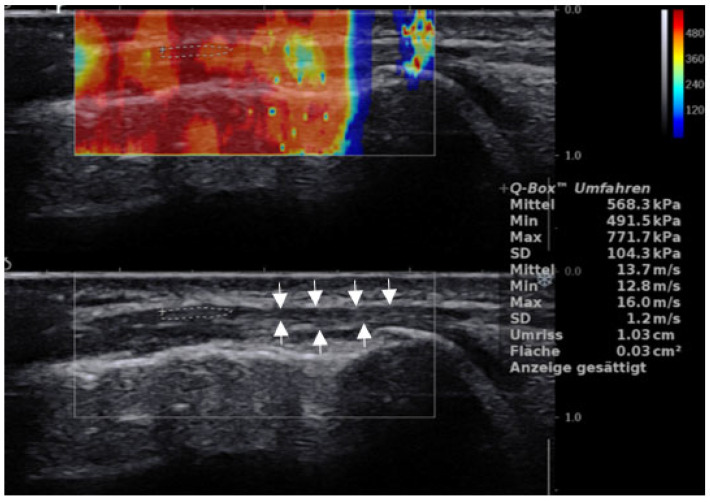
Longitudinal scan of the ITB. ROI shows measurement 2 cm proximal of the femoral condyle in a 29-year-old male patient with ITBS. Note: B mode US shows hypoechoic thickening and irregularity of the ITB (between arrows) with a mean of 13.7 m/s.

**Figure 2 jcm-11-03605-f002:**
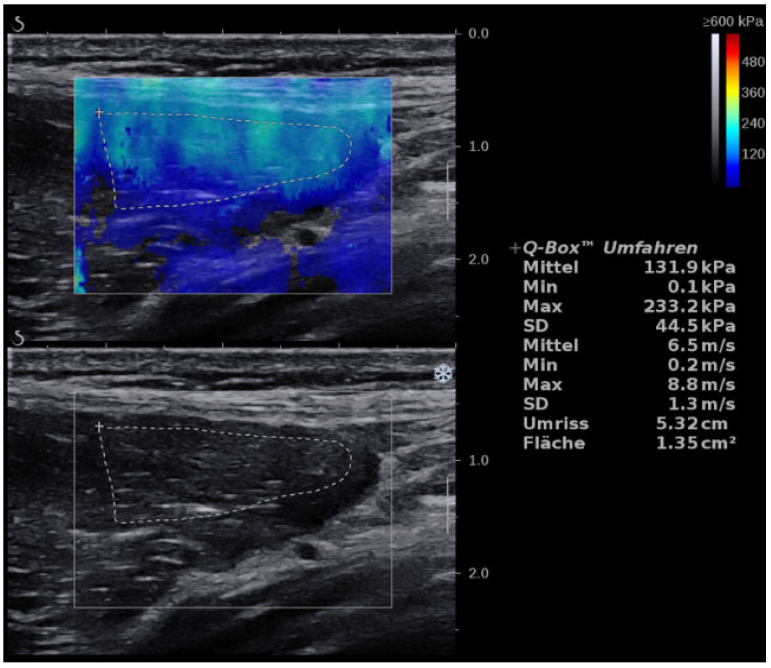
Longitudinal scan of the TFL muscle showing SWE and B-mode with a mean of 6.5 m/s. (the same patient with ITBS as in Figure 1).

**Figure 3 jcm-11-03605-f003:**
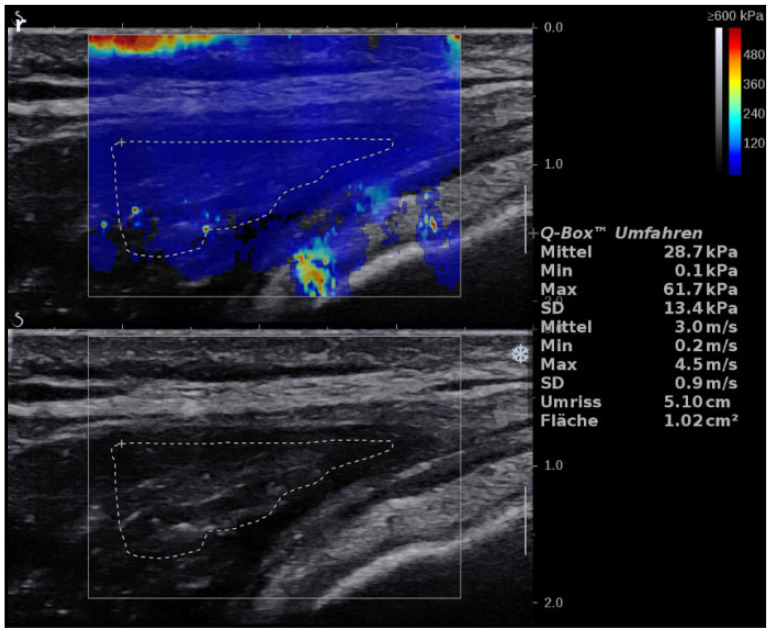
Axial scan of the GM muscle showing SWE and B-mode. (the same patient with ITBS as in Figure 1).

**Figure 4 jcm-11-03605-f004:**
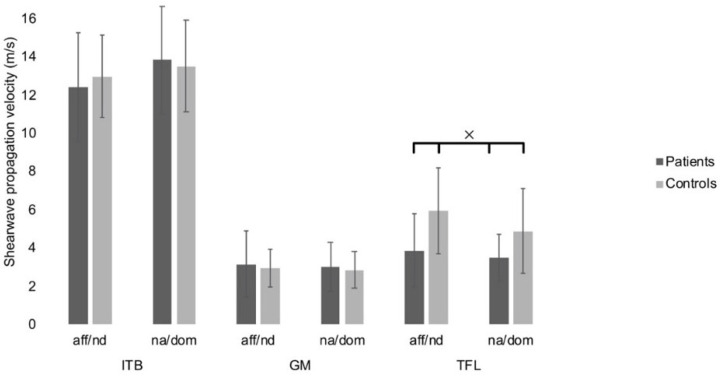
Baseline data of shear wave propagation velocity. Bars and error bars represent the means and standard deviations measured in the iliotibial band (ITB), gluteus maximus (GM) and tensor fasciae latae (TFL) muscles. Results are separately shown for the affected or non-dominant (aff/nd) and non-affected or dominant (na/dom) limbs, respectively. Note the significant difference between patients and healthy participants in the TFL (between cross).

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
