# Peer review of "Intraobserver Assessment of Shear Wave Elastography in Tensor Fasciae Latae and Gluteus Maximus Muscle: The Importance of the Hip Abductor Muscles in Runners Knee Compared to Healthy Controls"

_jcm, 2022, doi:10.3390/jcm11133605_

Round 1

Reviewer 1 Report

An interesting article, but for clearer results, I wish there were more patients
involved.

https://juniperpublishers.com/oroaj/OROAJ.MS.ID.555982.php

Author Response

An interesting article, but for clearer results, I wish there were more patients
involved.

https://juniperpublishers.com/oroaj/OROAJ.MS.ID.555982.php

Dear Reviewer, you are right – more patients would be better, but we think that we have enough power with our number of patients to be able to make a statement.
The article you sent is very interesting, we added it to our paper, thank you very much.

Reviewer 2 Report

Dears,

Thank you for the opportunity to read your manuscript. Some relevant considerations are presented below.

Overall:

The present manuscript has its merit and reports a stiffness assessment of ITB, gluteus maximus (GM) and tensor fasciae latae (TFL) muscles using shear wave elastography (SWE).

Specific comments:

Abstract

It is adequate partially for the purpose of the study protocol.

The aim presented in the abstract differs from that presented in the introduction item, what was the real aim of the study?

It is recommended to follow an existing abbreviation pattern in the literature for Iliotibial Band Syndrome (ITBS) and not ITB syndrome.

Keywords

The Gluteus minimus appears in the keywords, was this muscle evaluated in the study?

Introduction

Review the study aim description.

Methods

In the statistical analysis item, it was reported "The sample size was determined through a priori power analysis (É‘ = 0.05, 1-β = 0.8, dz = 1).” The authors performed a pilot study prior to sample size estimation? if yes, report.

Results

I respectfully interpret the reading of the result as confusing, as it does not follow a chronological order that is sought to be investigated in the aim of the study. I believe that the fact that my text has many corrections makes it difficult for the reader to analyze it. I suggest that it be rewritten in topics, to facilitate the interpretation of the results. Some acronyms that appear in the text are not clear about what is being evaluated.

Pag 5 line 165: What is CV?

Pag 4 line 165: ITB showed a mean of 13.24 ± 2.24 m/s (CV: 16.92%%); review %

Discussion

It is adequate for the purpose of the study, but it would be important to better describe the control group of the study.

Limitations

In the methodology item pag 2 line 83 it is reported "14 healthy volunteers 83 (median age 26.1 ± 5.2 / m: f = 7:7)", however in the limitations it is reported "A further limitation is, that we didn't examine a healthy runners group.” You mean you didn't compare runners with ITBS versus runners without ITBS, is that it? If yes, please better describe the control group in the methodology.

Figures

It is adequate for the purpose of the study.

Best Regards

Author Response

First of all, we would like to thank the reviewer for his valuable suggestions and hope that we have been able to improve the manuscript to his satisfaction. 

Dears,

Thank you for the opportunity to read your manuscript. Some relevant considerations are presented below.

Overall:

The present manuscript has its merit and reports a stiffness assessment of ITB, gluteus maximus (GM) and tensor fasciae latae (TFL) muscles using shear wave elastography (SWE).

Specific comments:

Abstract

It is adequate partially for the purpose of the study protocol.

The aim presented in the abstract differs from that presented in the introduction item, what was the real aim of the study?

The aim was to implement baseline values for stiffness assessments and prove reliability. We made this clear in the Abstract and Introduction.

It is recommended to follow an existing abbreviation pattern in the literature for Iliotibial Band Syndrome (ITBS) and not ITB syndrome.

We changed it accordingly throughout the manuscript.

Keywords

The Gluteus minimus appears in the keywords, was this muscle evaluated in the study?

No, this should be Gluteus maximus muscle, sorry for the misspelling – we changed it.

Introduction

Review the study aim description.

Thank you very much for your suggestion, we made it clear at the end of the Introduction.

Methods

In the statistical analysis item, it was reported "The sample size was determined through a priori power analysis (É‘ = 0.05, 1-β = 0.8, dz = 1).” The authors performed a pilot study prior to sample size estimation? if yes, report.

The sample size is based on previously published data of ITB stiffness (Tateuchi H, Shiratori S, Ichihashi N. The effect of three-dimensional postural change on shear elastic modulus of the iliotibial band. J Electromyogr Kinesiol. 2016;28:137-142. doi:10.1016/j.jelekin.2016.04.006.) and the assumption that ITB stiffness would be greater by more than one standard deviation in ITBS subjects.
We added this information in the paper.

Results

I respectfully interpret the reading of the result as confusing, as it does not follow a chronological order that is sought to be investigated in the aim of the study. I believe that the fact that my text has many corrections makes it difficult for the reader to analyze it. I suggest that it be rewritten in topics, to facilitate the interpretation of the results. Some acronyms that appear in the text are not clear about what is being evaluated.

You are right, for better reading we introduced 3 paragraphs on results of ITB, TFL and GM accordingly.
We have written out the abbreviations MDC and SEM, first time they appeared in the Methods section.

Pag 5 line 165: What is CV?

It means coefficient of variation. We added it to the text.

Pag 4 line 165: ITB showed a mean of 13.24 ± 2.24 m/s (CV: 16.92%%); review %

Thank you, we deleted the %.

Discussion

It is adequate for the purpose of the study, but it would be important to better describe the control group of the study.

Thank you very much, we tried to describe the control group better.

Limitations

In the methodology item pag 2 line 83 it is reported "14 healthy volunteers 83 (median age 26.1 ± 5.2 / m: f = 7:7)", however in the limitations it is reported "A further limitation is, that we didn't examine a healthy runners group.” You mean you didn't compare runners with ITBS versus runners without ITBS, is that it? If yes, please better describe the control group in the methodology.

Yes, that is the case! We made this clear in the limitations section.
„A further limitation is, that we didn´t examine a healthy runners group. This refers to healthy participants without ITBS symptoms with a self-reported weekly training volume of at least 20 km.“

Figures

It is adequate for the purpose of the study.

Thank you very much!

Reviewer 3 Report

The introduction does not mention that MRI is a well known imaging method in this issue, with clearly defined imaging findings. Also, some US findings have been described. MRI and US are though to be concordant.

Jiménez Díaz F, Gitto S, Sconfienza LM, Draghi F. Ultrasound of iliotibial band syndrome. J Ultrasound. 2020;23(3):379-385. doi:10.1007/s40477-020-00478-3

Flato R, Passanante GJ, Skalski MR, Patel DB, White EA, Matcuk GR Jr. The iliotibial tract: imaging, anatomy, injuries, and other pathology. Skeletal Radiol. 2017 May;46(5):605-622. doi: 10.1007/s00256-017-2604-y. Epub 2017 Feb 25. PMID: 28238018.

Patients and examiner position during US examination should be described, for others authors to be able to reproduce this technique. A figure could be added. 

As it is mentionned in the limitations section, l.292, thickness was not measured, it is clearly a limitation, as this finding could also be used as a diagnostic criterai given the lack of SWE significant difference between patients and controls. Also, MRI findings are not mentionned, maybe could they be compared to the distal ITB measurement? Was there bursitis or soft tissue edema that could influence SWE measurements?

The presence of a single radiologist is of concern.

Author Response

First of all, we would like to thank the reviewer for his valuable suggestions and hope that we have been able to improve the manuscript to his satisfaction. 

The introduction does not mention that MRI is a well known imaging method in this issue, with clearly defined imaging findings. Also, some US findings have been described. MRI and US are though to be concordant.

Jiménez Díaz F, Gitto S, Sconfienza LM, Draghi F. Ultrasound of iliotibial band syndrome. J Ultrasound. 2020;23(3):379-385. doi:10.1007/s40477-020-00478-3

Flato R, Passanante GJ, Skalski MR, Patel DB, White EA, Matcuk GR Jr. The iliotibial tract: imaging, anatomy, injuries, and other pathology. Skeletal Radiol. 2017 May;46(5):605-622. doi: 10.1007/s00256-017-2604-y. Epub 2017 Feb 25. PMID: 28238018.

We added a paragraph about MRI findings and included the papers provided by you – thank you very much.
„In addition, MRI is also a suitable method to evaluate the ITB and especially the surrounding structures for differential diagnosis. On MRI, the ITB is a hypointense, flat, linear structure in the lateral hip, thigh, and knee. In the absence of pathology, there should be no adjacent edema or significant intrasubstance signal changes. On MRI, the normal ITB will typically meas-ure a thickness of about 1 to 3 mm at the level of the lateral femoral epicondyle, con-sistent with sonographic findings (1, 13, 14). Nevertheless, thickening of the ITB has been reported inconsistently in MRI and US, therefore an independent imaging marker is desirable (15).“

Patients and examiner position during US examination should be described, for others authors to be able to reproduce this technique. A figure could be added. 

Thank you, this is a good point, we added a paragraph with explanation.
„To obtain SWE images, participants lay supine on an examination bed with their backs slightly raised and knees rested on a support cushion (hip angle 140°-150°, knee angle ~90°). US images were obtained in the sagittal and frontal plane, respectively, in both legs in three locations: proximally, above the tensor fasciae latae (2 cm proximal of the greater trochanter of the femur in the direction of the anterior superior iliac spine) and gluteus maximus muscles (4.5 cm proximal of the greater trochanter of the femur in the direction of the highest point on the iliac crest), and distally above the ITB (2 cm proximal of the lateral femoral epicondyle).“

As it is mentionned in the limitations section, l.292, thickness was not measured, it is clearly a limitation, as this finding could also be used as a diagnostic criterai given the lack of SWE significant difference between patients and controls. Also, MRI findings are not mentionned, maybe could they be compared to the distal ITB measurement? Was there bursitis or soft tissue edema that could influence SWE measurements?

Thank you for your suggestions! Indeed, it would be interesting to measure and correlate the thickness of the ITB, however, using SWE, we wanted to implement an independent parameter for diagnosis, because in the literature the thickness of the ITB is described very differently and controversially, especially in ITBS.
Regarding MRI, we used it mainly to rule out differential diagnoses. Unfortunately, we did not document any data regarding ITB. It would be indeed very interesting to correlate MRI with SWE and perhaps also thickness measurement in a further study. However, we do not expect bursitis or edema in the surrounding adipose tissue to influence SWE measurements, since we are measuring within the ITB.

The presence of a single radiologist is of concern.

Yes, this is of course a major limitation and we mentioned it in the limitations section.

Round 2

Reviewer 2 Report

Dears,

Thank you for the opportunity to read your manuscript. Some relevant considerations are presented below.

Overall:

The present manuscript has its merit and reports a stiffness assessment of ITB, gluteus maximus (GM) and tensor fasciae latae (TFL) muscles using shear wave elastography (SWE).

Specific comments:

Results

I suggest the following changes:

Pag 4 line 172: modify the “Results of the ITB” for “ITB”

Pag 4 line 177: “P>0.2” Here it was reported that "were statistically relevant", would not be p<0.02?

Pag 4 line 180: modify the “Results of the TFL” for “TFL”

Pag 4 line 185: modify the “Results of the GM” for “GM”

Best Regards

Author Response

Dear Reviewer, thank you very much for your valuable suggestions. We hope that we have been able to modify everything as you wished.

Pag 4 line 172: modify the “Results of the ITB” for “ITB”

We changed to "ITB".

Pag 4 line 177: “P>0.2” Here it was reported that "were statistically relevant", would not be p<0.02?

There it says: "Neither the differences between dominant/non-dominant as well as diseased/non-diseased legs were statistically relevant (P > 0.2)."
So it is okay like this - it was not significant.

Pag 4 line 180: modify the “Results of the TFL” for “TFL”

We changed to "TFL".

Pag 4 line 185: modify the “Results of the GM” for “GM”

We changed to "GM".